# Stem Cells: Engines of Plant Growth and Development

**DOI:** 10.3390/ijms241914889

**Published:** 2023-10-04

**Authors:** Liu Hong, Jennifer C. Fletcher

**Affiliations:** 1Plant Gene Expression Center, United States Department of Agriculture—Agricultural Research Service, Albany, CA 94710, USA; liuhong@berkeley.edu; 2Department of Plant and Microbial Biology, University of California Berkeley, Berkeley, CA 94720, USA

**Keywords:** meristem, stem cell, CLV3, WUS, KNOX, ARR, peptide, receptor, signal transduction, cytokinin, auxin, organogenesis, Arabidopsis

## Abstract

The development of both animals and plants relies on populations of pluripotent stem cells that provide the cellular raw materials for organ and tissue formation. Plant stem cell reservoirs are housed at the shoot and root tips in structures called meristems, with the shoot apical meristem (SAM) continuously producing aerial leaf, stem, and flower organs throughout the life cycle. Thus, the SAM acts as the engine of plant development and has unique structural and molecular features that allow it to balance self-renewal with differentiation and act as a constant source of new cells for organogenesis while simultaneously maintaining a stem cell reservoir for future organ formation. Studies have identified key roles for intercellular regulatory networks that establish and maintain meristem activity, including the KNOX transcription factor pathway and the CLV-WUS stem cell feedback loop. In addition, the plant hormones cytokinin and auxin act through their downstream signaling pathways in the SAM to integrate stem cell activity and organ initiation. This review discusses how the various regulatory pathways collectively orchestrate SAM function and touches on how their manipulation can alter stem cell activity to improve crop yield.

## 1. Introduction

Stem cells are cells with the unique capacity to replenish themselves and to act as source cells for the growth and development of multi-cellular organisms. Although multicellularity arose independently in animals and plants, both lineages evolved stem cells that have the ability to self-renew and to generate the various mature organs and tissues of the body. These stem cells are maintained by specialized microenvironments called niches that lie adjacent to the stem cells and support their function via localized signals. Given the essential nature of stem cells for both plant and animal development, their activity is under strict molecular control. Evidence indicates that the specific factors that regulate stem cell activity have diverged since the last common ancestor of the animal and plant lineages, and yet some of the underlying cellular and molecular mechanisms show notable similarity [1].

Like animal stem cells, plant stem cells initiate during embryogenesis. Animal stem cells are active predominantly during embryonic development, whereas plant stem cells exist in specialized reservoirs throughout the life cycle [2]. The lifelong maintenance of plant stem cell populations is thought to derive from the fact that plants are sessile and therefore modify their growth and development in response to changes in their immediate environment. Sustained stem cell activity enables plants to optimize root, leaf, stem, flower, and fruit formation under variable environmental conditions as well as to replace tissues lost to disease or predation. How stem cell reservoirs are established and maintained over the lifetime of a plant—which can extend several thousand years in species such as giant sequoias—has been the subject of intense research for nearly three decades and has led to a detailed understanding of a number of the key molecular mechanisms involved.

## 2. Meristem Structure and Function

Plant stem cells originate during embryogenesis [3] and are organized into structures called meristems, a term derived from the Greek word ‘merizein’ which means ‘to divide’. Two pluripotent stem cell populations are established as the embryo develops, forming a primary root and shoot apical meristem at the basal and apical poles of the mature embryo, respectively (Figure 1a). In the model plant *Arabidopsis thaliana* and other related species, the shoot apical meristem (SAM) is positioned between the two seed leaves, called cotyledons, and remains largely dormant until germination [4]. Becoming active in response to energy and light signals as the plantlet emerges from its seed coat [5], the SAM progresses through distinct developmental phases to generate the entire above-ground architecture of the plant. Following germination, the SAM acts as a vegetative meristem that produces leaves in a stereotypical pattern (Figure 1b) until various endogenous and environmental signals cooperatively cue the meristem to undergo the transition to flowering and reproductive development. During this phase, the stem elongates, secondary meristems form in the axils of leaves, and specialized floral meristems (FM) arise from both the primary and secondary SAMs (Figure 1c). These reproductive SAMs are referred to as inflorescence, or flower-bearing, meristems (IFMs), and their stem cell reservoirs can remain active indefinitely. Floral meristems, in contrast, contain transient stem cell pools that provide cells to form the flowers (Figure 1d), which consist of four organ types. The outer sepals and petals protect the developing bud and attract pollinators, whereas the inner stamens and carpels correspond to the male and female reproductive organs, respectively, with the latter maturing into fruits that house the seeds of the next generation.

The ability of the dome-shaped SAM at the growing shoot tip to perform its discrete functions of generating differentiated organs and tissues while simultaneously maintaining an active reservoir of pluripotent stem cells for future organogenesis is a result of its organization into distinct spatial domains. Cytological and histological analysis first revealed the existence of three radial domains [2] (Figure 2). The stem cells comprise a domain in the most apical, central region of the meristem called the central zone (CZ). As stem cells in the central zone slowly divide [6], some of their progeny cells are displaced towards the flanks of the meristem into a peripheral zone (PZ) of cells with reduced developmental potential [7] that surround the stem cells and divide more frequently, reminiscent of transit-amplifying cells in animal systems [8]. Cells in the PZ then gradually become incorporated either into organ primordia or into the internodal regions of the stem between the organs. Other stem cell descendants are displaced downward into the interior of the SAM, entering a domain called the rib zone (RZ) that makes up the bulk of the stem beneath the shoot apex. The uppermost cells of the RZ adjacent to the CZ play a crucial role in stem cell maintenance by acting as a stem cell niche or organizing center (OC). The OC sends important signals to the overlying central zone cells to specify their identity as stem cells and sustain their activity throughout growth and development, as detailed below.

The SAM is also organized into cell layers that are superimposed over the domain structure (Figure 2). In Arabidopsis, the L1 layer is the outermost layer of the SAM and gives rise to epidermal tissues. Directly beneath the L1 lies the sub-epidermal L2 layer that generates the mesodermal tissues as well as the germ cells of the pollen grains and ovules. A third group of cells denoting L3 produces the stem vasculature and the innermost cells of the leaves and floral organs. Cells in the L1 and L2 layers remain clonally distinct from one another by dividing perpendicular to the plane of the meristem, whereas the underlying L3 cells divide in all planes [9]. This arrangement allows surface growth to occur via the directional division of the L1 and L2 cells, while internal growth takes place through the division of the L3 cells in any orientation.

The different physiological features of the meristem domains and layers are reflected in their distinct molecular compositions. Transcriptomic evidence indicates that the different regions of the SAM have characteristic transcriptional profiles, as shown first by cell sorting and laser capture microdissection approaches in maize and Arabidopsis [10,11]. These studies found that the stem cells in the central zone are enriched in transcripts encoding proteins involved in DNA repair and chromatin modification. A high-resolution gene expression atlas of vegetative SAM and leaf domains generated using TRAP-seq later identified groups of region-enriched expressed genes, including 1656 in the CZ stem cell domain and 320 in the underlying OC [12]. Subsequent single-cell RNA-sequencing (scRNA-seq) analyses of shoot and floral meristems in several plant species uncovered multiple clusters of cells with distinct transcription profiles, indicating that meristems display a high degree of cell heterogeneity [6,13,14]. At the same time, many individual meristematic cells exhibit a continuum of intermediate identity states consistent with the occurrence of a continuous and dynamic differentiation process within the SAM [6,15]. Gene expression atlases also distinguished epidermal and vascular clusters corresponding to L1 and L3 SAM cells, respectively [16], and revealed gradations of cell cycle patterns from the L1 through the L3 layers [15]. Thus, these high-resolution methodologies begin to capture the complexity of meristem activity at the molecular level.

## 3. Establishing and Sustaining Plant Stem Cell Pluripotency

Mammalian stem cell integrity is controlled by master regulatory transcription factors such as OCT4, SOX2, and NANOG [17,18,19], as well as by signaling pathways mediated by the Wnt/b-catenin and TGF-b families, among others [20,21], which act to maintain cellular pluripotency in a cell lineage-specific fashion [20,22]. Notably, homologs of these and most other animal stem cell regulatory factors are absent from plant genomes, as plants have evolved distinct molecular factors for stem cell maintenance. Unlike animals, which generally evolved musculoskeletal systems to maintain their forms, the structural integrity of plants derives from the cell walls that encase each cell. This rigid framework prevents cell movement and therefore precludes lineage-dependent cell fate specification and cell migration. Rather, stem cell fate specification and SAM maintenance in plants are conferred by positional information [23,24] to keep the meristem at a relatively constant size. Such information is communicated by specific gene regulatory networks that have been the subject of intense investigation over the past several decades.

Interlocking regulatory networks involving transcription factors and phytohormones play key roles in sustaining stem cell activity in plant shoot and floral meristems. Arabidopsis *shoot meristemless (stm)* mutants, as the name implies, fail to establish a shoot apical meristem during embryogenesis, indicating that *STM* is required for SAM formation [25]. *STM* encodes a member of the class I KNOTTED-like homeobox (KNOX) gene family [26], which is part of the three amino acid loop extension (TALE) homeodomain superfamily present in all major eukaryotic lineages [27]. *STM* is expressed across the SAM [26] during the entire life cycle and is also required to sustain SAM activity throughout post-embryonic development [28], during which its activity overlaps with those of the closely related class I *KNOX* genes *BREVIPEDICELLUS (BP)* and *KNAT6* [29,30,31].

STM physically associates with various proteins to regulate meristem activity (Figure 3a). One is the SKI-INTERACTING PROTEIN (SKIP), a protein conserved in eukaryotes that regulates plant gene expression through both transcription regulation and mRNA splicing [32]. STM and SKIP can bind directly to the promoters of multiple STM target genes, and SKIP is proposed to act as an STM cofactor that recruits components of the transcriptional machinery, such as the RNA polymerase-associated factor 1 complex (Paf1c), to STM target genes to sustain SAM pluripotency [33]. Several members of the BELL family, another subclass of TALE homeodomain proteins [34], form heterodimers with STM to mediate high-affinity binding to target genes [35,36,37]. Three related *BELL* genes, *ARABIDOPSIS THALIANA HOMEOBOX 1 (ATH1)*, *PENNYWISE (PNY)*, and *POUND-FOOLISH (PNF)*, are together required for post-embryonic SAM activity in association with *STM* [38], with the BELL proteins mediating DNA binding and STM conferring transcription activation capacity [39]. STM lacks an effective nuclear localization sequence, and its import into the nucleus for transcription regulation requires heterodimerization with the BELL proteins [40]. Subcellular localization of BELL-KNOX complexes is mediated by the association of the nuclear export receptor CRM1, which shuttles proteins out of the nucleus, with conserved nuclear exclusion sequences in the BELL domain [38]. These BELL domain protein regions also interact with STM, indicating a mechanism whereby STM nuclear localization depends on its competition with CRM1 for association with BELL proteins [38].

The cell-to-cell movement of STM is important for maintaining meristem activity (Figure 3a), explaining why the transcription factor resides in the cytoplasm rather than in the nucleus by default. Intercellular STM trafficking is needed for the SAM to achieve its normal size, likely by allowing the protein to spread beyond the cells in which it is expressed [41]. Such is the case for KNOTTED1 (KN1), the maize ortholog of STM. *KN1* is expressed only in the interior layers of the maize SAM, but the mRNA and protein move to the overlying L1 layer [42] by trafficking through plasmodesmata [43]. These small cytoplasmic channels span plant cell walls to connect neighboring cells and facilitate the intercellular transport of macromolecules between them [44]. *STM* and *KN1* mRNA movement through plasmodesmata mediated by the RNA exosome subunit AtRRP44A has been shown to control stem cell-dependent processes in SAM cells [45]. KN1 protein movement requires the homeodomain [46] and is facilitated by the microtubule-associated movement protein binding protein 2C (MPB2C) [47] as well as by cytosolic chaperones [48]. Two additional factors, the multiple C2 domain and transmembrane region proteins FT INTERACTING PROTEIN3 (FTIP3) and FTIP4, directly regulate the movement of STM to the plasma membrane for trafficking in peripheral SAM cells and are essential for maintaining stem cell activity [49]. Tightly controlled trafficking of pluripotency master transcription factors and their mRNAs between cells is therefore a key mechanism for coordinating plant meristem function.

The main functions of STM are to suppress the differentiation of the meristem cells, preventing their incorporation into organ primordia [50], and to sustain their proliferative capacity [30]. One way in which STM and related class I KNOX proteins suppress meristem cell differentiation is by repressing gibberellic acid (GA) activity (Figure 3b). GA is a plant hormone, or phytohormone, that promotes differentiation by inducing cell expansion [51], and constitutive GA signaling reduces SAM activity [52]. STM and the tobacco class I KNOX protein NICOTIANA TABACUM HOMEOBOX 15 (NTH15) inhibit the expression of the GA biosynthesis gene *GA20 OXIDASE1 (GA20ox1)* in SAM cells [52,53], limiting GA levels within the meristem. STM and KN1 also induce the expression of several *GA2 OXIDASE* genes encoding enzymes that inactivate bioactive GAs [54,55]. Class I KNOX proteins therefore modulate GA levels in the SAM by both inhibiting biosynthesis and promoting catabolism. In addition, STM is necessary to repress the expression of genes that mediate organ specification and patterning, including the *ASYMMETRIC LEAVES (AS1)* and *AS2* transcriptional regulatory genes [56,57,58] as well as the *TEOSINTE BRANCHED/CYCLOIDEA/PCF1 (TCP)* transcription factor genes *TCP3* and *TCP4* [59], to prevent organ initiation within the meristem proper.

STM and related class I KNOX proteins sustain stem cell pluripotency at least in part through their regulation of cytokinin activity (Figure 3b). Cytokinins (CKs) are phytohormones derived from N^6^-substituted adenines that act systemically to promote cell proliferation [60]. CKs are synthesized by the *ISOPENTENYL TRANSFERASE (IPT)* and *LONELY GUY (LOG)* genes, several of which are expressed in the SAM [61,62]. Plants carrying mutations in *LOG* genes form smaller than normal meristems [63,64], whereas Arabidopsis *altered meristem program 1 (amp1)* mutants with elevated CK levels have vastly enlarged SAMs [65]. Cytokinin, therefore, plays a crucial role in supporting SAM function. Inducible activation of *STM* leads to the rapid up-regulation of several *IPT* genes, primarily *IPT7*, and a corresponding increase in the levels of several active cytokinins [66], and CK application can partially rescue the *stm* meristem phenotype [55]. STM promotes stem cell proliferation by acting through CKs to modulate the cell cycle, inducing the expression of *CYCLIN D3 (CYCD3)* cell cycle regulatory genes to simulate cell division [30]. CYCD3 proteins then bind and activate cyclin-dependent kinases and direct the entry of cells into S-phase for DNA replication [67]. Collectively, these data indicate that class I KNOX factors sustain stem cell pluripotency by controlling hormone activity networks to maintain low levels of GA and high levels of CK within the meristem [55,66].

Analysis of the gene regulatory networks of STM, KN1, and the rice ortholog ORYZA SATIVA HOMEOBOX1 (OSH1) proteins reveals that class I KNOX targets not only components of the GA and CK pathways but also those of other hormones such as auxin and brassinosteroids (BRs) [59,68,69]. Auxin is a versatile developmental regulatory hormone that influences cell division, elongation, and differentiation and is important for organ positioning and outgrowth. KN1 directly binds to genes involved in multiple aspects of the auxin pathway, including biosynthesis, transport, catabolism, perception, and signal transduction [68]. Although the significance of this association is not understood, it may be related to the fact that auxin is maintained at a low level within the SAM interior, whereas high auxin concentrations are required to position organ primordia on the meristem flanks [70,71]. BRs are growth-promoting hormones involved in regulating the division, elongation, and differentiation of various plant cell types [72]. In rice, inducible *OSH1* overexpression causes BR deficiency, whereas *osh1* loss-of-function leads to BR overproduction phenotypes [69]. Consistently, OSH1 binds to multiple BR pathway genes, and its induction results in the rapid up-regulation of three BR catabolic genes, *CYTOCHROME P450 734A2 (CYP734A2), CYP734A4*, and *CYP734A6*, that prevent the premature differentiation of the SAM [69]. Thus, the BR catabolism pathway is an additional important target of class 1 KNOX factors to maintain stem cell pluripotency.

In plants as well as in animals, repression of pluripotency programs is important for organ formation. Although *STM* is expressed throughout the SAM, it is down-regulated in incipient vegetative and reproductive organ primordia arising on the meristem flanks [26], and its repression promotes the initiation of floral primordia [73]. During the vegetative phase, the AS1 and AS2 transcriptional regulatory proteins, in combination with TCP transcription factors [74], form a heteromeric complex that directly represses *KNOX* gene expression early during the differentiation of newly arising leaf cells [31,75,76]. In initiating flowers, the silencing of *STM* expression occurs epigenetically via histone methylation mediated by POLYCOMB REPRESSIVE COMPLEX (PRC1) and PRC2 [77], as well as histone deacetylation mediated by direct binding of the histone deacetylase HDA19 and a YABBY/ARF repressive complex to the *STM* locus [73]. In these ways, key pluripotency genes are stably silenced during cell differentiation (Figure 3b).

## 4. Regulation of Plant Stem Cell Maintenance

A complex gene regulatory network called the CLAVATA (CLV)-WUSCHEL (WUS) pathway is required to maintain stem cell homeostasis during post-embryonic plant development. Three *CLV* loci, *CLV1, CLV2*, and *CLV3*, encode components of a signal transduction cascade that functions to limit stem cell accumulation within shoot and floral meristems, as loss-of-function mutations in any of these genes result in the progressive enlargement of the stem cell reservoirs, leading to the formation of thick fasciated stems and the production of flowers with extra floral organs [78,79,80]. *CLV3* encodes a small secreted polypeptide that is expressed exclusively in the stem cells of all above-ground meristems, beginning during embryogenesis and continuing throughout the life cycle [81]. It is a founding member of the CLAVATA3/EMBRYO SURROUNDING REGION-RELATED (CLE) family of peptide ligands that is conserved across the land plant lineage [82,83]. *CLE* mRNAs are translated into pre-proteins that consist of an amino-terminal signal peptide, a conserved 14-amino acid carboxyl-terminal CLE domain, and an intervening variable domain [82]. The signal peptide is sufficient to direct the pre-proteins through the cell secretory pathway to the extracellular space [84,85] and is needed for their function in plants [86]. The CLE pre-proteins are proteolytically processed by secreted proteases [87,88,89,90] to generate biologically active 12 or 13 amino acid peptides consisting of the CLE domain [91,92].

Mature CLV3 peptides are localized to the extracellular space and undergo post-translational modification to obtain full biological activity. The first two of the three proline residues in the CLV3 peptide, Pro^4^ and Pro^7^, are modified to hydroxyproline [91], and the Hyp^7^ residue is additionally decorated with three L-arabinose residues. The latter modifications enhance CLV3 peptide activity in Arabidopsis [92] and in tomato and are mediated by three sequentially-acting arabinosyltransferases [93]. Similar to other CLE peptides, CLV3 likely forms a horseshoe-shaped kink around the central Gly^6^ and Hyp^7^ residues that is important for receptor recognition [94], with the L-arabinose residues at Hyp^7^ inducing a conformational alteration in the carboxyl-terminal half of the CLV3 peptide and enhancing receptor binding affinity [95].

The extracellular CLV3 ligand is perceived by a suite of plasma membrane-localized receptors (Figure 4). *CLV1* was the first component of the signal transduction pathway to be characterized [78] and encodes one of a large family of leucine-rich repeat receptor-like kinases (LRR-RLKs) that is expressed in the interior layers of the SAM beneath the stem cell domain [96]. CLV1 proteins are capable of auto-phosphorylation [97,98] and form homodimers at the plasma membrane [99,100] that directly bind CLV3 peptides via their extracellular LRR domains [101]. CLV3 perception by CLV1 triggers an intercellular signal transduction cascade that functions to limit stem cell accumulation by restricting the expression domain of the *WUS* gene [102], which encodes a transcription factor that maintains stem cell identity [103,104]. At the cellular level, CLV3 ligand binding sequesters the CLV1 receptors within plasma membrane micro-domains [100] and activates their cytosolic kinase domains, which leads to TPLATE complex-dependent receptor endocytosis [105] and trafficking to the lytic vacuole for degradation [106]. This ligand-mediated receptor internalization process attenuates CLV3 signaling within the SAM to prevent premature meristem termination [100,105,107].

The CLV1 LRR-RLK has three paralogs encoded by the *BARELY ANY MERISTEM1 (BAM1), BAM2*, and *BAM3* genes that are primarily expressed on the periphery of the SAM [108,109]. As their name implies, the three *BAM* genes act together to promote stem cell maintenance [108]. However, mutations in either *BAM1* or *BAM2* enhance the *clv1* null mutant phenotype, indicating that BAM1 and BAM2 function redundantly with CLV1 to perceive the CLV3 signal [110]. Indeed, like CLV1, BAM1 and BAM2 can directly bind CLV3 peptides [111,112]. Evidence shows that CLV1 activity in the RZ represses *BAM* gene transcription, independently of its role in regulating *WUS* transcription, such that in *clv1* mutants, ectopic *BAM* expression in the interior of the SAM partially compensates for the absence of *CLV1* [109,113].

Two additional receptor complexes transduce the CLV3 signal in parallel with CLV1 (Figure 4). RECEPTOR-LIKE PROTEIN KINASE2 (RPK2) is an LRR receptor-like kinase of a distinct sub-family from that of the CLV1/BAM RLKs [114] that is present throughout the SAM and is required for SAM maintenance [115]. RPK2 proteins homodimerize but function independently of CLV1 and CLV2 [115]. Photoaffinity labeling experiments fail to detect binding of the RPK2 extracellular domain to CLV3 peptides [111], suggesting that the RPK2 complex may require a co-receptor for ligand perception. An attractive candidate is BAM1, which can physically associate with RPK2 to modulate cell proliferation in the root apical meristem [116]. A third receptor complex is comprised of the CLV2 and CORYNE (CRN) proteins. CLV2 is a receptor-like protein with an extracellular LRR domain, a transmembrane region, and a short cytoplasmic domain [117] that produces a slightly enlarged meristem phenotype when mutated [80]. CLV2 forms a heteromeric complex at the plasma membrane with the CRN pseudokinase, a membrane-associated protein with an inactive cytoplasmic kinase domain that likely acts as a CLV2 co-receptor [118,119,120]. CLV2 and CRN interact in the endoplasmic reticulum via the CRN transmembrane domain and require one another for localization to the plasma membrane [99]. Genetic evidence indicates that the CLV2/CRN complex is functionally separate from the CLV1 and RPK2 complexes and mediates distinct signaling outputs in plants [113], although the presence of CLV3 peptides triggers the formation of CLV1/CLV2/CRN multimers that cluster in membrane subdomains [100]. Evidence is inconclusive as to whether CLV2 can directly bind the CLV3 peptide [111,112]. Thus, the CLV2/CRN complex may require additional partners, such as CLV1, to perceive and transduce the CLV3 signal. Similar to *RPK2, CLV2* and *CRN* are expressed throughout the SAM [117,118], and the regulation of *CRN* transcription by histone methylation is necessary for meristem maintenance. The type II arginine methyltransferase SKB1/PRMT5 mediates the systemic demethylation of H4R3 to repress *CRN* expression, which limits *CLV3* and *WUS* transcription levels to modulate stem cell activity [121].

Recent studies revealed that a quartet of LRR-RLK proteins with short extracellular domains, called the CLAVATA3 INSENSITIVE RECEPTOR KINASES (CIKs), act as co-receptors with the CLV1, RPK2, and CLV2/CRN receptor complexes [122]. *cik1234* enlarged meristems are similar in size to *clv3* and *clv1 clv2 rpk2* meristems, consistent with the CIK proteins participating in all three receptor-mediated pathways. Phosphorylation of the CIK proteins by CLV1 and the BAM receptors is an early event in CLV3 signal transduction that is required to maintain stem cell homeostasis in the SAM [122].

Although the complete CLV3-mediated signaling pathway from the apoplast to the nucleus has yet to be elucidated, a number of intermediate components have been identified that transduce the CLV3 signal within the cell. A clade of receptor-like cytoplasmic kinases (RLCKs), PBS1-LIKE34 (PBL34), PBL35, and PBL36, act downstream of CLV1 and BAM1/3 to mediate CLV3 peptide intracellular signaling. These proteins physically associate with CLV1, BAM1/3, and the CIK LRR-RLKs and are phosphorylated by CLV1 and BAM1 [123]. PBL34 can also phosphorylate BAM3 *in vitro* [124]. The PBL proteins appear to act independently of CLV2 and RPK2, suggesting that they may be components of a CLV1-specific intracellular signaling pathway for SAM maintenance [123]. Conversely, the MAZZA (MAZ) membrane-associated RLCK can physically interact with CLV1, BAM1/3, CRN, and CIK2, suggesting a potential role for MAZ in integrating signals from various CLV3 receptors [125]. However, although *MAZ* is expressed throughout inflorescence and floral meristems, single-mutant *maz* phenotypes are limited to stomata and root development, and a role for *MAZ* in SAM maintenance is yet to be identified.

Protein phosphatases are among the earliest-characterized CLV3 signaling intermediates. In the late 1990s, the type 2C protein phosphatase KINASE ASSOCIATED PROTEIN PHOSPHATASE (KAPP) and a Rho GTPase-related protein called ROP were shown to associate with CLV1 receptor kinases [126], with KAPP dephosphorylating the CLV1 kinase domain and acting as a negative regulator of CLV1-mediated signaling [97,98]. The *POLTERGEIST (POL)* gene was first identified as a partial suppressor of the *clv1* shoot and floral meristem phenotype [127]. *POL* and *POLTERGEIST-LIKE1 (PLL1)* encode dual-acylated plasma membrane-bound phosphatase 2C proteins that bind to and are catalytically activated by select phospholipids [128,129]. Both genes act downstream of CLV1 receptor signaling and are essential for stem cell maintenance throughout development [130,131,132]. Recent evidence indicates that the RLCK PBL34 directly phosphorylates POL and PLL1 to diminish their ability to attenuate CLV1/BAM signaling [124]. The paradigm is therefore that in the absence of CLV3 ligand, the PLL family phosphatases dampen LRR-RLK signaling by inhibiting the auto- and trans-phosphorylation of the receptors, whereas perception of CLV3 by the LRR-RLKs leads to RLCK co-receptor recruitment to phosphorylate the PLL family phosphatases and promote their dissociation from the complex to facilitate intracellular signal transduction [124].

Heterotrimeric G proteins are also important intracellular CLV3 signaling components. Present in both animals and plants, the core heterotrimeric G protein complex consists of alpha (α), beta (β), and gamma (γ) subunits that bind GTP and link extracellular signals to downstream effectors that induce cellular responses [133]. The *DENSE AND ERECT PANICLE1 (DEP1)* gene encoding a Gγ subunit regulates SAM size and functions as a major QTL for grain yield in rice [134,135]. Maize *compact plant2 (ct2)* mutants with defects in the canonical Gα subunit gene have enlarged meristems resembling those of Arabidopsis *clv* mutants, and CT2 is required to transmit the CLV signal within the cell [136]. The CT2 Gα subunit likely functions in a complex with the Gβ subunit [137] and also physically associates with FASCIATED EAR2 (FEA2), the maize ortholog of the CLV2 receptor-like protein [136]. Similar to the situation in rice, genes encoding the CT2/Gα subunit, the Gβ subunit, and several non-canonical EXTRA LARGE G-PROTEIN (XLG) Gα subunits all regulate SAM activity and influence maize yield traits [137,138]. CLV3 signaling in Arabidopsis also involves both heterotrimeric G protein α and Gβ subunits, although the single Gβ subunit, ARABIDOPSIS G-PROTEIN BETA SUBUNIT (AGB1), physically interacts with RPK2 rather than with CLV2 or CLV1 [139,140]. Heterotrimeric G protein activity is thus a conserved mechanism for intracellular stem cell signal transduction downstream of the CLV receptor complexes. 

Receptor-like kinases involved in regulating plant developmental processes commonly transmit intracellular signals through mitogen-activated protein kinase (MAPK) cascades [141,142]. MAPK cascades are evolutionarily conserved signaling intermediates involving the sequential action of MAPK kinases and MAPKs, all of which are represented by multiple related proteins in Arabidopsis [143]. MAPK activity has been implicated in signaling downstream of the CLV receptors, with MPK6 protein kinase activity being induced in response to an exogenous CLV3 stimulus [144]. A subsequent study confirmed the phosphorylation of MPK6 following CLV3 peptide application and suggested that CLV3-induced phosphorylation of both MPK6 and MPK3 is mediated by the CLV1 and BAM1 receptors [145]. Because *mpk3* and *mpk6* null mutants are embryo-lethal [142], the precise contribution of these MAPKs to maintaining SAM integrity remains to be fully defined.

The key biologically relevant target of the CLV signaling pathway in Arabidopsis is the *WUS* transcription factor gene. WUS is the founding member of the *WUSCHEL-LIKE HOMEOBOX (WOX)* family of homeobox genes that are distantly related to *STM* and other *KNOX* genes. In addition to their characteristic DNA-binding homeodomain, several Arabidopsis WOX proteins, including WUS, also contain an acidic domain, an 8-amino-acid WUS-box motif, and a carboxyl-terminal EAR repressor motif [146]. Loss-of-function *wus* mutant plants form a defective primary SAM that produces only a few leaves before the stem cell reservoir is consumed [103]. Additional SAMs repetitively initiate during vegetative and reproductive development but terminate prematurely, indicating that *WUS* is required for the maintenance of stem cell fate [103,104]. *WUS* expression initiates in the 16-cell embryo and is restricted to the OC domain directly beneath and adjacent to the stem cells in shoot and floral meristems [104]. Interestingly, two short sequence motifs within the *WUS* proximal promoter are sufficient to confer proper spatial and temporal expression of the gene in the stem cell niche [147]. *WUS* expression expands both laterally into the PZ and also upwards into the L2 cell layer in *clv3* mutant plants, whereas constitutive *CLV3* expression abolishes *WUS* transcription and terminates stem cell activity [102]. These data demonstrate that CLV3 signaling is required to limit *WUS* function by restricting its expression to the OC in the interior of the SAM. In addition, CLV3 signaling attenuates the nuclear export of WUS protein from the nucleus into the cytosol (Figure 4), which limits its movement from the L3 cells into the outer cell layers of the SAM and also prevents its lateral movement and spread [148].

Additional CLE peptides contribute to Arabidopsis stem cell maintenance and *WUS* regulation along with CLV3, as first indicated by the observation that *clv1 bam1 bam2 bam3* plants form larger SAMs than *clv3* plants [113]. The simultaneous mutation in the *clv3* background of 11 *CLE* genes that displayed repressive activity in peptide assays indeed generated a more severe phenotype than *clv3* null mutants [149]. However, the *dodeca-cle* phenotype was still weaker than that of the receptor quadruple mutant, implicating the contribution of yet more *CLE* genes. Systemic analysis of Arabidopsis *CLE* gene expression patterns revealed that the *CLE16, CLE17*, and *CLE27* genes, which were not mutated in the *dodeca-cle* plants, are expressed within or adjacent to the SAM [150]. Although these three *CLE* genes are dispensable for SAM function when mutated singly [151], CLE16 and CLE17 can partially compensate for CLV3 in restricting stem cell accumulation by signaling through the BAM receptor kinases to negatively regulate the *WUS* expression domain [152]. Many *CLE* genes therefore act together to buffer stem cell homeostasis in Arabidopsis shoot and floral meristems. A passive compensation mechanism among *CLE* paralogs also functions in maize, whereas an active mechanism involving up-regulation of *SlCLE9* transcription to counterbalance the loss of *SlCLV3* activity has been described in tomato [149].

Several additional CLE peptides orchestrate meristem homeostasis independently of CLV3. The *CLE40* gene is expressed in the PZ of the Arabidopsis SAM in a complementary pattern to *CLV3* and signals through the peripheral BAM1 receptor kinase to maintain *WUS* expression in the stem cell niche, while WUS in turn represses *CLE40* transcription in the CZ and OC [153]. This CLE40 pathway is proposed to operate antagonistically to the CLV3 pathway to replenish cells on the flanks of the meristem that become incorporated into organ primordia. CLE signaling from the SAM periphery also occurs in maize, where the ZmFON2-LIKE CLE PROTEIN1 (ZmFCP1) peptide is produced in developing organ primordia [154]. ZmFON2 peptides act through the CLV2 ortholog FEA2 and an additional LRR receptor-like protein called FEA3 to suppress *ZmWUS1* transcription specifically in cells below the OC [154,155]. These data indicate that not only multipotent cells but also cells undergoing differentiation communicate with the stem cell niche to maintain meristem integrity. Therefore, CLE peptide pathways that originate from three distinct regions of the shoot apex—the CZ of stem cells, the PZ, and the organ primordia—all serve to coordinate meristem homeostasis by converging on the regulation of *WUS* expression.

*CLE* genes are found throughout the land plant lineage and in some plant-parasitic nematodes [83,156,157], and regulation of stem cell activity appears to be an ancestral feature of the family. The liverwort *Marchantia polymorpha* genome contains two *CLE* genes, *MpCLE1* and *MpCLE2*, the latter of which encodes a peptide of the CLV3 subfamily [158]. MpCLE2 peptide signaling through the MpCLV1 and MpCIK receptors regulates apical meristem activity in the liverwort gametophyte, suggesting that the CLV3-CLV1-CIK module is evolutionarily conserved in land plants [159,160]. However, this module inhibits cell differentiation and positively regulates stem cell number in liverwort apical meristems [160], reminiscent of the CLE40-BAM module in Arabidopsis [153]. Gene clustering analysis of CLV ligand and receptor genes indicates that the CLV3 and CLV1 clusters comprise genes found only in flowering plants, whereas the CLE40 and BAM clusters contain genes found both in flowering plants and in non-flowering vascular plants [156,161]. These observations have led to the proposal that the stem cell-promoting activity of the CLE40-BAM module may be the ancestral function acquired early during the evolution of land plants and that the stem cell-restricting CLV3-CLV1 module is a derived state that arose through gene duplication in the common ancestor of flowering plants [161].

## 5. Maintenance of Plant Stem Cell Activity

WUS is a central hub in the network that regulates above-ground plant stem cell activity. WUS activity is not only necessary to maintain SAM function throughout the life cycle [103] but is also sufficient to convert organ cells to stem cell fate [162] and to confer shoot stem cell identity on cells in the root [163]. Transient manipulation of *WUS* expression coupled with live imaging showed that elevating *WUS* levels increases PZ cell division rates and induces expansion of the CZ via the re-specification of PZ cells as stem cells [164]. In addition, WUS maintains stem cell activity through its interactions with cytokinin signaling pathways and mediates the balance between multipotency and differentiation in the PZ by affecting the responsiveness of the PZ cells to auxin (see below). The molecular mechanisms through which WUS achieves these cell-autonomous and non-cell-autonomous effects on stem cell homeostasis and meristem integrity have been the subject of extensive research over the past few decades.

One critical function of WUS in meristem maintenance is the regulation of *CLV3* expression in the stem cells (Figure 5a). WUS activity not only confers stem cell fate but is also sufficient to induce *CLV3* expression in CZ cells [162]. Although *WUS* expression is restricted to the OC, WUS protein is also detected in the overlying CZ, where it is required to maintain SAM activity [165,166]. Similar to STM, WUS protein moves between cells through plasmodesmata [165], mediated by the C-terminal 63 amino acids of the protein [167]. This movement generates a gradient of high WUS accumulation in the OC and lower accumulation in the L2 and L1 layers [165]. Formation of the gradient involves the nuclear-cytoplasmic partitioning of WUS protein, as increased nuclear targeting reduces its movement into the outer cell layers [165]. This partitioning is mediated by the WUS-box, which is required for nuclear retention, and the EAR-like domain, which functions as a nuclear export sequence via its physical association with EXPORTIN proteins [162,167]. Cytokinin signaling also plays a role in sustaining the WUS gradient, acting through the WUS acidic domain and WUS-box to stabilize the protein in the OC [168]. In the CZ, where CK levels are lower, the WUS-box functions as a degradation sequence that destabilizes the protein and reduces its accumulation in the stem cells.

WUS directly binds to multiple cis elements at the *CLV3* locus in both the CZ and OC and regulates its expression in a concentration-dependent fashion (Figure 5b,c) [165,169]. High levels of WUS protein in the OC lead to the DNA-dependent formation of homodimers [170] that repress *CLV3* transcription, whereas low levels of WUS protein in the CZ result in the prevalence of monomers that induce *CLV3* transcription [171]. In turn, CLV3 signaling from the CZ into the OC either directly or indirectly promotes WUS protein stability, potentially by inhibiting its lateral diffusion into adjacent cells [148], allowing the formation of nuclear WUS homodimers that repress *CLV3* transcription in the deeper layers of the SAM. These data are consistent with the CLV-WUS pathway forming a negative feedback loop [102,162] in which WUS activity in the CZ induces *CLV3* expression while the size of the WUS domain of activity is limited post-translationally by CLV3 signaling to sustain shoot and floral meristem homeostasis.

Additional factors also contribute to the spatial regulation of *CLV3* by WUS in the different regions of the SAM (Figure 5a). A small family of *HAIRY MERISTEM (HAM)* genes encoding GRAS-domain transcription factors are required for meristem maintenance [172,173,174] and are expressed in overlapping domains in the PZ and RZ [173,175,176]. The HAM1/2/3 proteins act in a non-cell-autonomous fashion to regulate *CLV3* expression in the CZ [172,173,175]. The HAM1 and HAM2 proteins accumulate to high levels only within the L3 cells [177] due to their negative regulation by a mobile miR171 signal, the expression of which is directly induced by the epidermal-specific ARABIDOPSIS THALIANA MERISTEM LAYER1 (ATML1) and PROTODERMAL FACTOR2 (PDF2) transcription factors [178]. HAM1 and HAM2 form heterodimers with WUS in the OC and act as WUS co-factors to regulate WUS downstream targets [176]. Association with HAM proteins prevents WUS from activating *CLV3* in the OC, confining *CLV3* transcription to the overlying cells and thus delineating the apical-basal position of the *CLV3* expression domain [177] in a reciprocal pattern to that of the HAM proteingradient [178]. Further evidence for the importance of WUS homo- and heterodimerization in regulating *CLV3* transcription comes from a study showing that in developing floral meristems, the C2H2 zinc finger transcription factor KNUCKLES (KNU) inhibits the binding of WUS to the *CLV3* locus by disrupting WUS homodimer and WUS-HAM heterodimer formation, which induces stem cell termination [179].

STM is also necessary for meristem maintenance. It is required to sustain *WUS* expression and to prevent the incorporation of *WUS*-expressing cells into organ primordia [50]. In addition, *STM* and *KNAT6* are epistatic to *CLV3* during SAM development [180], and the formation of a WUS-STM heteromeric complex increases WUS binding to the *CLV3* promoter and enhances its transcriptional activity [181]. This suggests that the movement of low levels of WUS proteins from the OC into the CZ, where HAM proteins are absent, enables the formation of WUS-STM complexes that induce *CLV3* expression in the stem cells (Figure 5a). These data have led to a model wherein SAM cell fate is regulated by a combination of transcription factors acting in overlapping domains [181]. In the stem cells, STM and WUS act synergistically to promote *CLV3* expression, whereas in the OC, this function is blocked by the presence of the HAM proteins. In the PZ, WUS is absent, so STM may act to prevent premature cell differentiation but does not induce *CLV3* transcription. Finally, in initiating primordia, the absence of both WUS and STM allows the differentiation programs required for organ formation to proceed.

WUS acts as a bifunctional transcription factor that can both activate and repress gene expression [182]. The repressive effect of WUS is mediated via its interaction with transcriptional co-repressor proteins. The WUS protein physically associates with the TOPLESS (TPL) co-repressor as well as several related TPR proteins to repress gene transcription [183]. The interaction occurs via the conserved carboxyl-terminal domain, with the WUS-box and EAR motif being essential for the interaction between the two proteins [184,185]. TPL mediates auxin-dependent transcriptional repression during embryogenesis and associates with the HDA19 histone deacetylase to form a transcription repression complex [186]. Although genetic evidence has not yet confirmed a functional role for Arabidopsis TPL family proteins in SAM maintenance, likely due to their redundancy and involvement in many core biological processes [187,188], the maize *TPL*-like gene *RAMOSA1 ENHANCER LOCUS2 (REL2)* is required for IFM maintenance and axillary meristem initiation [189]. Collectively, these studies suggest a molecular mechanism whereby recruitment by WUS of TPL-HDA19 repressor complexes confers the chromatin-mediated stable repression of downstream gene transcription required for SAM maintenance.

The WUS protein binds to DNA as a dimer via its homeodomain to three distinct motifs, with a strong preference for TGAA repeat sequences over G-box-like elements and TAAT motifs [170]. Genome-wide transcriptome analysis identified 675 WUS-responsive genes in the Arabidopsis genome, of which 159 were determined to be potential direct targets based on chromatin immunoprecipitation (ChIP) experiments [190]. Genes with roles in development, cell division, and hormone signaling were overrepresented among the direct WUS targets, including genes involved in cytokinin signaling as well as those involved in auxin biosynthesis, signaling, and transport [190]. A subsequent study showed that WUS directly binds to and represses the transcription of genes expressed in differentiating cells on the meristem flanks, including *KANADI1 (KAN1), KAN2, AS2*, and *YABBY3 (YAB3)* [191] (Figure 5a). Because these genes regulate lateral organ development, it is thought that WUS-mediated transcriptional repression of these genes is necessary to prevent the premature differentiation of stem cell progenitors [191].

## 6. Hormone Signaling in Meristem Maintenance

Both the CLV-WUS signaling pathway and the cytokinin signaling pathway are crucial for SAM maintenance, and their molecular networks are intricately connected (Figure 6a). Active CKs produced by the IPT and LOG enzymes are recognized by two-component receptors encoded by several *ARABIDOPSIS HISTIDINE KINASE (AHK)* genes [192,193]. CK perception initiates receptor autophosphorylation and an intracellular phosphorelay cascade that results in the phosphorylation and activation of type-B *ARABIDOPSIS RESPONSE REGULATOR (ARR)* genes that induce cytokinin-responsive gene transcription [194]. Among the genes activated by the type-B ARR proteins are type-A *ARR* genes, which encode proteins structurally similar to type-B ARRs but lack a DNA binding domain [195]. The type-A ARR proteins inhibit type-B ARR protein function, acting as repressors of CK signaling in a negative feedback loop [196]. All elements of this system intertwine with CLV-WUS-mediated signaling in the SAM.

WUS regulates several cytokinin response genes to enhance CK activity in the SAM (Figure 6a). The protein directly represses the transcription of *ARR7* and three other type-A *ARR* genes that negatively regulate CK signaling, and constitutive activation of *ARR7* mimics the *wus* mutant phenotype [197]. Conversely, down-regulation of type-A *ARR* genes enhances SAM function and promotes shoot regeneration [198,199,200]. An important feature of WUS activity is therefore to limit the ability of type-A *ARR* genes to dampen cytokinin signaling, thereby sustaining robust shoot apical meristem activity.

Cytokinin response pathways also impact the CLV-mediated stem cell maintenance module by regulating WUS activity. CK treatment perceived by the AHK2 and AHK4 receptors increases *WUS* transcription and enhances SAM size through both *CLV*-dependent and *CLV*-independent mechanisms [201]. Analysis of CK receptor mutants indicated that CK signaling is also required for WUS protein stability [168]. The Arabidopsis *LOG4* gene is expressed in the L1 layer of the SAM and floral meristem, and as CKs are diffusible molecules, this spatial restriction of CK biosynthesis in the SAM is postulated to form a diffusion gradient downward towards the interior cells [61]. The *AHK4* CK receptor gene expression domain lies a few cell layers away, in the RZ, with the upper half of the domain overlapping that of *WUS* [61,201]. These observations have been integrated into a computational model indicating that this epidermally-derived CK gradient acts together with the CLV signaling pathway to position the *WUS* expression domain within the stem cell niche [61,62]. This molecular mechanism involving coordinate CK and CLV signaling modulates *WUS* expression scales to various meristem geometries [62] and may also mediate the adaptation of SAM size to nutrient availability [202]. Additionally, type-B ARR transcription factors that function as downstream positive effectors of cytokinin signaling directly regulate *WUS* at the transcriptional level. Several type-B ARR proteins directly bind and induce *WUS* transcription in a CK-dependent manner by associating with short core type-B ARR binding sites in the *WUS* promoter [203,204,205]. These type-B *ARR* genes are required for axillary meristem formation and shoot regeneration [203,204] and generate greatly enlarged SAMs when constitutively active [205]. The induction of *WUS* expression by type-B ARR proteins that induce the CK response, along with the repression by *WUS* of the activity of type-A *ARR* genes that dampen the CK response, results in a positive feedback loop that promotes SAM function and helps sustain the *WUS* expression domain in the OC region where CK levels are highest (Figure 6a).

Cytokinins are positive regulators of cell proliferation that promote cell cycle progression and activate mitotic cell division. In the SAM, CK promotes cell progression into the S-phase via the cyclin-dependent kinase A/D-type cyclin (CYCD) pathway by activating the expression of three *CYCD3* genes that maintain the SAM stem cells in an undifferentiated state and prevent endoreduplication [67,206]. CK also promotes the nuclear shuttling of the MYB domain transcription factor MYB3R4 that is required for CK-induced cell division in the SAM [207]. At the G2-M transition, nuclear localization of MYB3R4 associated with a transient spike in CK concentration enables its physical interaction with MYB3R1 to activate a transcriptional cascade that drives mitosis and cytokinesis [207]. Finally, a recent study indicates that the activity of CK, WUS, and mechanical stress combines in a region-specific fashion to regulate cell division plane orientations and cell expansion patterns within the SAM [208]. These studies provide new insights into the precise orchestration of SAM activity at the cellular level.

The plant hormone auxin is a key determinant of plant developmental patterning, including the precise positioning of organ primordia around the flanks of the SAM. Auxin is a derivative of the amino acid tryptophan, and the major active auxin, Indole-3-Acetic Acid (IAA), rapidly diffuses between cells. As such, the accumulation of auxin at concentrations sufficient to orchestrate developmental patterning is achieved through the directional transport of auxin via PIN-FORMED (PIN) family polar auxin transport proteins [209]. Cellular responses to auxin are mediated by Aux/IAA and AUXIN RESPONSE FACTOR (ARF) proteins [210]. Aux/IAA proteins are short-lived nuclear proteins that repress auxin responses by forming heterodimers with the ARF transcription factors and interfering with their DNA binding when auxin levels are low. Higher concentrations of auxin are perceived by the TIR1 receptor, a component of a ubiquitin ligase complex. Binding of auxin to the TIR1 ubiquitin ligase complex promotes its association with the Aux/IAA proteins, leading to their degradation and thereby permitting the ARF proteins to homodimerize and alter downstream gene transcription.

Auxin-mediated control of organ patterning within the SAM is a dynamic process [211,212]. In Arabidopsis, auxin is synthesized in the developing leaves and is transported by the PIN1 protein upward through the L1 cell layer into the flanks of the SAM [70,213]. There, auxin accumulates at high levels (maxima) concurrently at several sites, generating a stereotypical spiral organ initiation pattern [212] and also feeding back through an auxin transport switch to maintain SAM homeostasis [214]. PIN1-mediated auxin transport is required for organogenesis, as *pin1* mutants form IFMs that lack organs [215], and auxin application to the flanks of *pin1* meristems induces organ initiation in a position- and concentration-dependent fashion [70]. An early link between auxin and the classical meristem maintenance pathways came from the observation that transient down-regulation of *WUS* causes a larger than normal proportion of PZ cells to become incorporated into organ primordia by elevating their responsiveness to auxin [164]. This reveals a non-cell-autonomous role for WUS not only in regulating stem cell fate in the CZ but also in modulating the balance between differentiating and non-differentiating cells in the PZ by influencing their auxin response [164].

The interplay between the CLV-WUS pathway and hormone signaling in SAM regulation is fine-tuned by the region-specific activity of several classes of transcription factors. The *HECATE1 (HEC1)* gene encodes a basic helix-loop-helix (bHLH) TF that is expressed in the CZ and PZ but is directly repressed by WUS in the OC [216]. *HEC1* together with the closely related *HEC2* and *HEC3* genes enhance SAM activity by inhibiting the transition from stem cell to peripheral cell fate and by antagonizing OC cell activity [216,217]. Consistently, HEC1 and WUS oppositely regulate a large proportion of their shared target genes, including *CLV3* as well as the *ARR7* and *ARR15* type-A cytokinin response regulators, with WUS repressing and HEC1 inducing the expression of the latter two genes (Figure 6b). HEC1 thus interferes with CK signaling and indirectly reduces *WUS* transcript levels, which is proposed to occur non-cell-autonomously in the OC via movement of the ARR proteins [216]. Conversely, HEC activity is necessary and sufficient to promote CK signaling in the stem cell domain, increasing SAM size [218]. The HEC TFs also modulate local auxin signaling in the SAM. HEC1 directly regulates the transcription of the *PIN1* and *PIN3* genes [219] and reduces the expression of the key auxin response gene *MONOPTEROS (MP)*, also called *ARF5* [217]. Moreover, the HEC proteins physically associate with MP, forming a complex thought to negatively affect auxin signaling in the PZ by destabilizing the positive feedback loop between MP and PIN1 [218,220]. In this way, the HEC TFs promote CK signaling in the CZ while dampening auxin signaling in the PZ to shape the timing of cell differentiation across the SAM [218]. Another auxin response gene, *ETTIN/ARF3*, is induced predominantly at the periphery of the SAM, where it coordinates the placement of initiating organ primordia around the meristem flanks [221]. In addition, ARF3 proteins are translocated into the OC cells, where they directly repress *WUS* transcription and CK activity to limit meristem size in a non-cell-autonomous fashion [221].

Auxin also accumulates in the CZ [212], where the stem cells require a low level of the hormone for their maintenance [222]. However, the CZ shows reduced auxin sensitivity compared to the PZ [223], and the stem cells are resistant to auxin-mediated differentiation signals [222]. Consistently, *MP* is expressed at lower levels in CZ cells than in PZ cells, with the protein forming a gradient from the PZ into the CZ [200]. MP-mediated auxin signaling nevertheless plays an important role in stem cell regulation. MP directly represses transcription of the *DORNROSCHEN/ENHANCER OF SHOOT REGENERATION1 (DRN/ESR1)* gene encoding an AP2/ERF family TF that is expressed in the stem cell domain [224]. DRN activity in the stem cells induces *CLV3* expression, and together with its homolog DRNL, it limits SAM cell accumulation and overall meristem size [225]. In addition, MP directly represses *ARR7* and *ARR15* expression to promote CK signaling in the SAM [200], illustrating crosstalk between the auxin and cytokinin pathways in meristem maintenance. MP-mediated auxin signaling therefore indirectly negatively regulates *CLV3* expression by repressing *DRN* and type-A *ARR* gene expression, which in turn sustains WUS activity and stem cell function (Figure 6b) [225].

The low auxin signaling state of stem cells in the Arabidopsis SAM is dependent on stem cell fate and WUS activity [222]. At the same time, WUS is required in the stem cells to directly repress the expression of *MP* as well as many other auxin-responsive genes, largely through histone deacetylation. These results have led to the proposal that WUS functions as a rheostat that permits low levels of auxin signaling in the stem cells sufficient to maintain their activity while preventing the amplification of the signal to levels that would drive their differentiation [222]. By limiting the transcription of auxin response genes in the CZ through chromatin modifications, WUS is hypothesized to restrict organogenesis to the PZ, where auxin signaling can occur [226]. Overall, the interconnectedness of the CLV-WUS feedback loop with the cytokinin and auxin signaling pathways provides long-term robustness to shoot stem cell homeostasis.

## 7. Conclusions and Perspectives

The unique, life-long developmental plasticity of plants is fueled by the continuous activity of stem cell reservoirs at their growing tips. The importance of shoot stem cell maintenance to plant life is reflected in the complexity of the underlying molecular mechanism that governs their activity, an intricate network composed of intercellular phytohormone and peptide signaling pathways as well as gradients of small regulatory RNAs and mobile transcription factors. The use of cutting-edge technologies is giving us a better understanding of the dynamic nature of the meristem maintenance process and how cells located in different regions of the SAM acquire their distinct fates. The discovery that CLE-mediated stem cell regulation is an ancestral feature of land plants and is associated with the emergence of three-dimensional body plans underscores the relevance of stem cell research across the land plant lineage. New theoretical and technical breakthroughs will continue to provide novel insights into this fundamental biological process.

Beyond the insights gained into the organizing principles of plant stem cell biology, evidence has also emerged that key genes in the stem cell signaling pathways have been selected during crop domestication to increase productivity. Increases in fruit compartment, or locule, number associated with cultivated tomato species are largely due to two quantitative trait loci (QTL), *locule number (lc)*, and *fasciated (fas)*. These QTL result from regulatory mutations at the *WUS* and *CLV3* loci, respectively, that enhance fruit size and were independently selected during the tomato domestication process [93,227]. Similarly, increases in the number of rows of kernels produced by ears of corn (maize) contributed to the domestication of that crop [228,229] via the selection of QTL that map to two different maize CLV receptors [230,231]. In these ways, the historical selection of natural variation in meristem maintenance genes has led to the improvement of major crops for human benefit.

The advent of genome editing has enabled the targeting of SAM regulatory genes to accelerate the domestication of new crop species as well as improve the efficiency of plant regeneration. Two studies demonstrated the use of genome editing to generate weak alleles of *CLV3* that increased fruit size as part of the *de novo* domestication of wild tomato species [232,233], while engineering mutations in the *CLV1* gene improved fruit size in the orphan crop groundcherry, a relative of tomato and pepper [234]. Furthermore, components of the auxin and cytokinin signaling pathways, as well as WUS and various CLE peptides, play key roles in regulating shoot regeneration [235,236], which is important for the propagation and transformation of many plant species via tissue culture. These advances show that manipulating shoot stem cell regulatory networks is a promising strategy for improving plant productivity, one that will become increasingly refined and effective as we continue to investigate these remarkable engines of plant development.

## Figures and Tables

**Figure 1 ijms-24-14889-f001:**
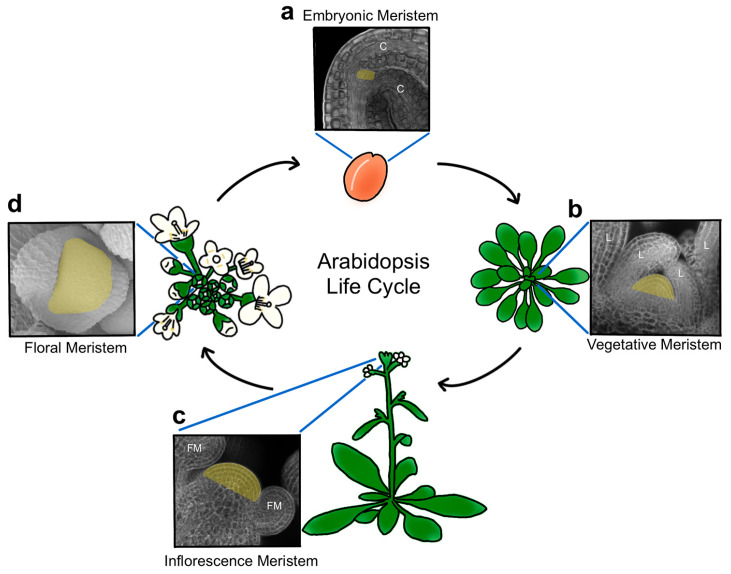
Phases of meristem activity during the plant life cycle. (**a**) The embryonic shoot apical meristem is a small dome of cells located at the base of the cotyledons (C) in the mature Arabidopsis embryo. (**b**) The vegetative meristem produces rosette leaves (L) from its flanks. (**c**) The inflorescence meristem generates floral meristems (FM) around its circumference. (**d**) The floral meristem contains a transient stem cell reservoir that provides the cells for the four organs of the flower. At this stage, only the outer sepal organ primordia are visible in the inset. The meristem cells are falsely colored yellow.

**Figure 2 ijms-24-14889-f002:**
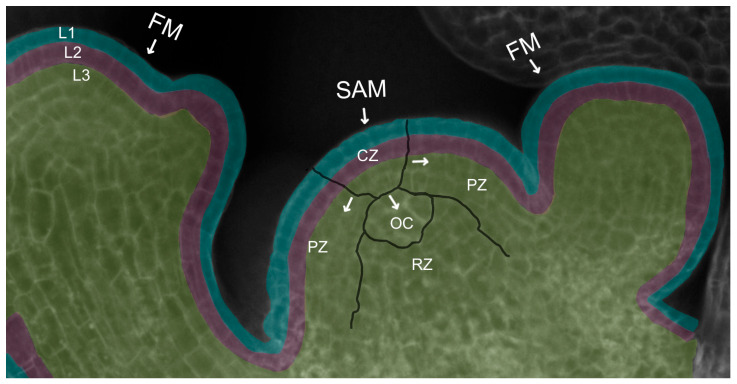
Organization of the Arabidopsis shoot apical meristem. Arrangement of the SAM into the central zone (CZ), peripheral zone (PZ), organizing center (OC), and rib zone (RZ). Colors depict the arrangement of the SAM and adjacent FMs into three cell layers: the epidermal L1 layer (blue), the sub-epidermal L2 layer (purple), and the interior L3 cell layers (green). Arrows denote the transition of cells from the CZ outward into the PZ or downward into the OC.

**Figure 3 ijms-24-14889-f003:**
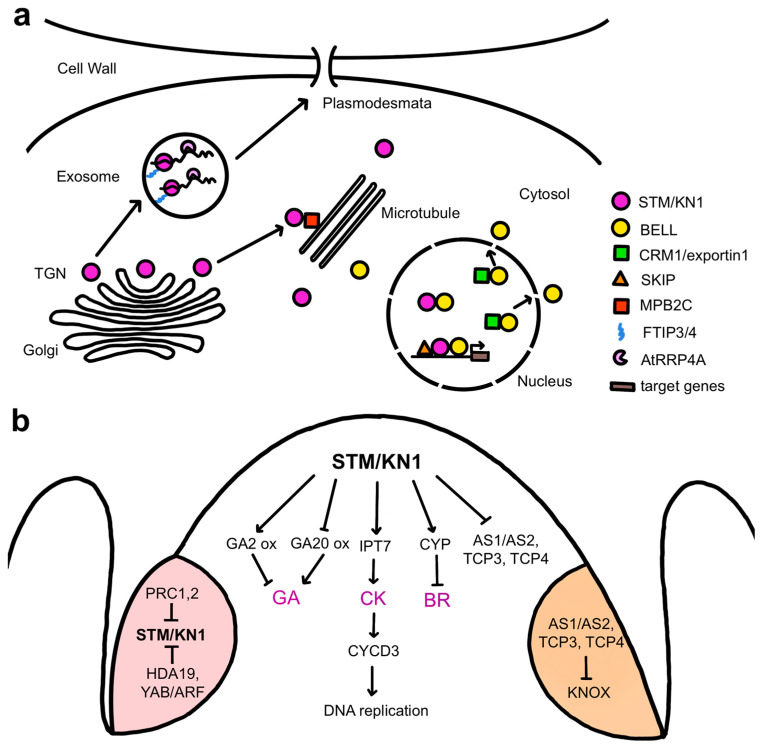
Sub-cellular localization and gene regulatory networks of the class I KNOX factors STM/KN1 within the SAM. (**a**) Sub-cellular localization of STM and/or KN1 and regulated movement of KN1 between cells via plasmodesmata. Arrows denote protein movement. (**b**) Key downstream targets of STM and/or KN1 in the SAM and negative regulation of *STM* and/or *KN1* in initiating organ primordia. A leaf primordium is shaded orange, and an initiating flower is shaded pink. Arrows denote positive regulation, and bars denote negative regulation of gene expression.

**Figure 4 ijms-24-14889-f004:**
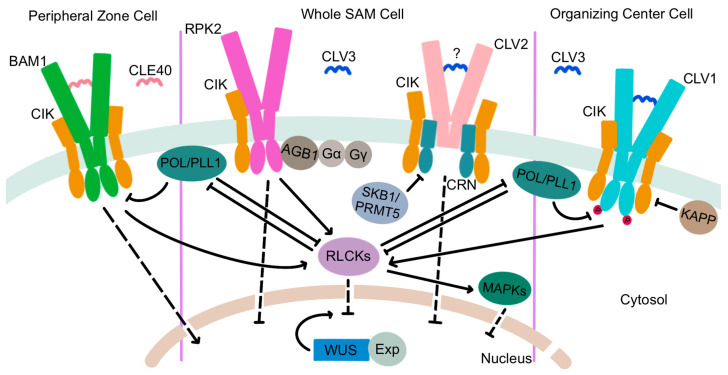
Components of the CLE signaling pathways in different regions of the SAM. The RPK2 and CLV2/CRN receptor complexes are required in all SAM cells to perceive the CLV3 signal. The CLV1 receptor complex directly binds to the CLV3 peptide in the OC cells. These receptor complexes are proposed to inhibit nuclear export of WUS protein by EXPORTIN proteins (Exp) and restrict its diffusion into adjacent cells. The BAM1 receptor complex is required in the PZ cells to perceive the CLE40 signal and induce *WUS* activity. Solid lines denote direct regulatory interactions, and dashed lines denote indirect regulatory interactions. Arrows denote positive regulation, and bars denote negative regulation of gene expression or protein activity. Red circles denote phosphorylation. The question mark indicates it is unclear whether CLV2 directly binds the CLV3 peptide.

**Figure 5 ijms-24-14889-f005:**
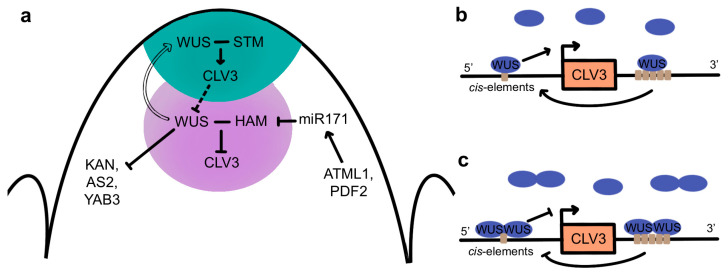
The CLV-WUS feedback loop in the SAM. (**a**) Regulatory inputs and outputs of the CLV-WUS meristem maintenance network across the various SAM layers and domains. The CZ is shaded in blue and the OC in purple. (**b**) Induction of *CLV3* expression in the stem cell domain via the binding of WUS monomers at low concentrations to cis-elements in the promoter and 3′ region (**c**). Repression of *CLV3* expression in the organizing center via the binding of WUS dimers at high concentrations to cis-elements in the promoter and 3′ region. Solid lines denote direct regulatory interactions, and dashed lines denote indirect regulatory interactions. Arrows denote positive regulation, and bars denote negative regulation of gene expression. Short bars (-) denote protein-protein interactions. The transparent arrow denotes the movement of WUS protein from the OC to the CZ.

**Figure 6 ijms-24-14889-f006:**
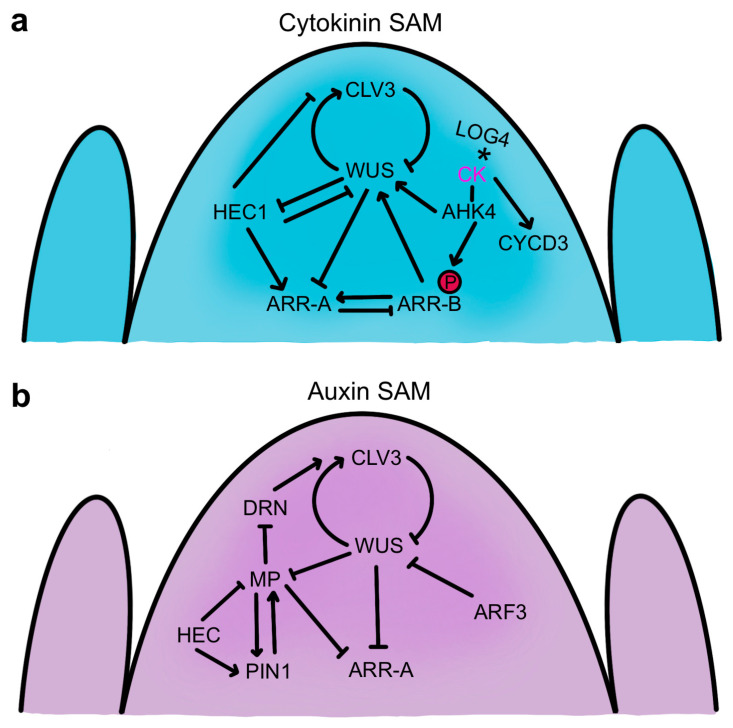
Hormone regulation of SAM activity. (**a**) Interplay between the cytokinin signaling network and the CLV-WUS stem cell feedback loop. An asterisk denotes cytokinin activation by the LOG4 enzyme. A red circle denotes phosphorylation. (**b**) Interplay between the auxin signaling network and the CLV-WUS stem cell feedback loop. Arrows denote positive regulation, and bars denote negative regulation of gene expression.

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
