# Peer review of "Stem Cells: Engines of Plant Growth and Development"

_ijms, 2023, doi:10.3390/ijms241914889_

Round 1
Reviewer 1 Report
In this review, Hong and Fletcher have captured their deep understanding about plant stem cell biology. A very positive aspect of the review is that it often compares plants with animals in terms of pluripotent cell biology, which makes the review appropriate for this Special Issue.
The review is comprehensive, updated and reallly well written, so my recommendations to authors are more related to issues of form than content. Regarding the latter, it would be useful to briefly define the role of auxin phytohormones in plant biology in line 254, the first time that these hormones are quoted in the text, as the authors do with BRs, GAs and CKs. It´s true, that the role of auxins in meristem maintenance is discussed in section 6, but that is quite far.
Regarding style, I recommend some changes to authors in order to make the article more comprehensible. The document lacks italics, perhaps because of some problem with its formatting after uploading it to the journal platform. A thorough review is needed for all terms that should be italicized: species names (such as Arabidopsis thaliana in line 54), gene names (for example, STM in line 157) and mutant names (such as shoot meristemless (stm) in line 156. Please, revise all over the paper.
Although most of the time when a gene is cited for the first time the full name appears, which is very positive for the reader, this is not always the case. Some examples of this are given in line 191 (KN1, KNOTTED1), 222 (NTH15, NICOTIANA TABACUM HOMEOBOX 15), 245 (CYCD3, CYCLIN D3), 264 (CYP, CYTOCHROME P450), 590 (ATML1 and PDF2), etc.
Please, unify the way to refer to the rib zone (RZ) of the SAM wich appears as RZ in line 94, and as rib meristem in lines 586 and 686 (better RZ).
Next, a couple of recommendations to improve the illustration of the article:
Figure 1 is nice and and suitable for not plant-readers, but can be improved. In 1a indicate with two “c” both cotyledons. In 1b use a color to mark the vegetative meristem, as in 1a, or indicate it on the picture with “vm”. Do the same using the letter “l” for the leaves. In 1c indicate the inflorescence meristem with “im” and with “fm” the flower meristems around it. Also in 1c, it would be nice to mark the secondary meristems in the axils of cauline leaves. Incorporate in the figure legend all this lettering.
In figure 4 would be covenient to use a different color for CLE40, since the same shape and color has been used for depicting CLV3. In the same figure, I assume that the blue ellipse attached to WUS in the nucleus is an EXPORTIN but it is not indicated in the legend. Please, clarify in figure legend or in lines 470-471.
Finally, some minor concerns to be addressed:
Line 171. Add “complex” after Polymerase-Associated Factor1.
Line 241. Remove 7 in IPT7.
Line 511. Indicate the whole name of Marchantia polymorpha in italics.
Line 846. Remove the whole line.
Author Response
The review is comprehensive, updated and reallly well written, so my recommendations to authors are more related to issues of form than content. Regarding the latter, it would be useful to briefly define the role of auxin phytohormones in plant biology in line 254, the first time that these hormones are quoted in the text, as the authors do with BRs, GAs and CKs. It´s true, that the role of auxins in meristem maintenance is discussed in section 6, but that is quite far.
- Thank you for the positive comments. We have added a sentence that describes the general roles of auxin on lines 255-257.
Regarding style, I recommend some changes to authors in order to make the article more comprehensible. The document lacks italics, perhaps because of some problem with its formatting after uploading it to the journal platform. A thorough review is needed for all terms that should be italicized: species names (such as Arabidopsis thaliana in line 54), gene names (for example, STM in line 157) and mutant names (such as shoot meristemless (stm) in line 156. Please, revise all over the paper.
- The original manuscript submission in Word format had the correct italics throughout the paper. We have italicized the appropriate terms throughout the revised IJMS-formatted version.
Although most of the time when a gene is cited for the first time the full name appears, which is very positive for the reader, this is not always the case. Some examples of this are given in line 191 (KN1, KNOTTED1), 222 (NTH15, NICOTIANA TABACUM HOMEOBOX 15), 245 (CYCD3, CYCLIN D3), 264 (CYP, CYTOCHROME P450), 590 (ATML1 and PDF2), etc.
- We have made these changes.
Please, unify the way to refer to the rib zone (RZ) of the SAM wich appears as RZ in line 94, and as rib meristem in lines 586 and 686 (better RZ).
- We have changed rib meristem to RZ as requested for consistency.
Next, a couple of recommendations to improve the illustration of the article:
Figure 1 is nice and and suitable for not plant-readers, but can be improved. In 1a indicate with two “c” both cotyledons. In 1b use a color to mark the vegetative meristem, as in 1a, or indicate it on the picture with “vm”. Do the same using the letter “l” for the leaves. In 1c indicate the inflorescence meristem with “im” and with “fm” the flower meristems around it. Also in 1c, it would be nice to mark the secondary meristems in the axils of cauline leaves. Incorporate in the figure legend all this lettering.
- We have made these requested changes, although because this article does not focus on secondary meristems we prefer not to label them in Figure 1c for simplicity.
In figure 4 would be covenient to use a different color for CLE40, since the same shape and color has been used for depicting CLV3. In the same figure, I assume that the blue ellipse attached to WUS in the nucleus is an EXPORTIN but it is not indicated in the legend. Please, clarify in figure legend or in lines 470-471.
- We have made these changes.
Finally, some minor concerns to be addressed:
Line 171. Add “complex” after Polymerase-Associated Factor1.
Line 241. Remove 7 in IPT7.
Line 511. Indicate the whole name of Marchantia polymorpha in italics.
Line 846. Remove the whole line.
- We have made all of these changes.
Reviewer 2 Report
This is a very well organized and well written review covering a large body of complex literature on shoot meristem development. As expected much of it is based studies on Arabidopsis, authors also make a sincere effort to bring in relevant literature from other plant systems. The figures are nice and very easy to follow. I have listed few points that pertain to providing details on certain observations which will bring forth different points of view. I suggest that these must be included to balance the review and they may also guide future studies.
1. Line 37 Authors state that “Animals stem cells are active predominantly during embryonic development, whereas plant stem cells”. I am not sure this is a correct statement. There are specialized stem cells such as germ line stem cells, Hematopoietic stem cells etc are maintained during post-embryonic stages.
2. Line 103 Authors use wording “mesodermal tissues”. This sounds like “animal terminology”. May be “Sub-epidermal” could be better.
3. Line 469 Authors state that “In addition, CLV3 signaling attenuates the nuclear export of WUS protein from the nucleus into the cytosol (Figure 4), which limits its movement from the L3 cells into the outer cell layers of the SAM [148]”.
Not just limits its movement from L3 into outer layers but also prevents lateral movement/spread. This enhanced lateral movement/spread can explain SAM overproliferation observed in clv3 null mutants.
4. Line 544 Authors state that “Like STM, WUS protein moves between cells through plasmodesmata, mediated by its homeodomain [165, 166]”.
Yes there is evidence that WUS moves through plasmodesmata but homeodomain is not necessary for its movement because it has been shown that the last 63 amino acid stretch which excludes the N-terminal homeodomain can move (Rodriguez et al., PNAS 2016, p. E6307-E6315.). This must be corrected.
5. Line 587 Authors state that “The HAM1 and HAM2 proteins form a concentration gradient within the SAM [177], accumulating to high levels within the L3 cells but to low levels in the L1 and L2 layers”
However, there is no evidence that HAM1 and HAM2 proteins that are synthesized in the deeper L3 layers reach the L1 layer to form protein gradients. See the work presented in (Han et al., Front. Plant Sci., Volume 11 2020. https://doi.org/10.3389/fpls.2020.541968) where HAM proteins are not detected in the L1 layer. It is an important point which shows that these proteins must act non-cell autonomously to effect CLV3 expression in the CZ. This notion of non-cell autonomous function is also suggested by original study that discovered HAM in petunia (Stuurman et al., genes and dev 2002, 16(17):2213-8) where differentiating cells thought to influence meristem maintenance. It is important to highlight this strong evidence for their non-cell autonomous function which is useful for future studies in explaining of how the WUS-HAM heterodimers that form in the inner layers influence CLV3 expression and stem cells in outer cell layers. I can also provide one more instance of discrepancy that exists in the literature which is worth highlighting to bring clarity to their non cell autonomous function. Schulze, S., et al. Plant Journal showed that only ham1;ham2;ham3 triple mutants develop extremely flattened reorganized meristems and the ham1;ham2 double mutants were closer to the wild type suggesting again that the PZ-expressed Ham3 contributes non cell autonomously. A similar conclusion on triple mutant phenotype was reached by another study Engstrom et al., Plant Physiol 2011 Feb;155(2):735-50. Highlighting these aspects is necessary to get anyway closer to the understanding the function of WUS-HAM heterodimers.
6. In paragraph starting with line 602 authors discuss WUS-STM heterodimers directly activating the CLV3 expression based on Su et al., PNAS 2020.
WUS-STM heterodimers bind to upstream elements TAAT located at -1080 and STM by itself binds TGACA located at (–797 bp to –793 bp). The double mutants have been shown to abolish binding and such mutant promoter failed completely in rescuing clv3 null mutants showing their critical role. However, authors overlook mentioning two important studies. First, -1080 was identified as a WUS-binding element by Yadav et. Al., 2011 Gene and Dev, and the deletion of this element did not influence CLV3 expression. Moreover, Muller at al., Plant Cell, Volume 18, Issue 5, May 2006, Pages 1188–1198, carried out sequential deletions of the promoter. The deletions that leave behind only upstream 500bps (-500) was sufficient to fully rescue clv3 null mutants. The functionality was only compromised upon deletion of the 3’ region. These two studies contradict the critical requirement of the STM binding sites located upstream of the 500bps. These differences should be mentioned.
7. In paragraph starting with line 649, authors discuss CK receptor requirement for activating WUS transcription. They should also mention a more direct experiment showing no effect on the WUS expression domain in triple mutants of the cytokinin receptors, rather the WUS protein stability is compromised in these mutants [Snipes et al., Plos Genetics 2018, 14(4)].

Author Response
- Line 37 Authors state that “Animals stem cells are active predominantly during embryonic development, whereas plant stem cells”. I am not sure this is a correct statement. There are specialized stem cells such as germ line stem cells, Hematopoietic stem cells etc are maintained during post-embryonic stages.
- We appreciate the reviewer’s comments and agree that some animal stem cells are maintained during post-embryonic stages. However, the bulk of animal stem cell activity occurs during embryogenesis to generate the tissues and organs of the organism. Therefore, we think the use of the phrase “active predominantly during embryonic development” is accurate.
- Line 103 Authors use wording “mesodermal tissues”. This sounds like “animal terminology”. May be “Sub-epidermal” could be better.
- We use the term “sub-epidermal” earlier in the sentence to refer to the L2 layer, and use “mesodermal tissues” to make the tissue association clear to readers more familiar with the animal stem cell literature.
- Line 469 Authors state that “In addition, CLV3 signaling attenuates the nuclear export of WUS protein from the nucleus into the cytosol (Figure 4), which limits its movement from the L3 cells into the outer cell layers of the SAM [148]”.
Not just limits its movement from L3 into outer layers but also prevents lateral movement/spread. This enhanced lateral movement/spread can explain SAM overproliferation observed in clv3 null mutants.
- We have made this addition.
- Line 544 Authors state that “Like STM, WUS protein moves between cells through plasmodesmata, mediated by its homeodomain [165, 166]”.
Yes there is evidence that WUS moves through plasmodesmata but homeodomain is not necessary for its movement because it has been shown that the last 63 amino acid stretch which excludes the N-terminal homeodomain can move (Rodriguez et al., PNAS 2016, p. E6307-E6315.). This must be corrected.
- Thank you for the correction, we have made this change.
- Line 587 Authors state that “The HAM1 and HAM2 proteins form a concentration gradient within the SAM [177], accumulating to high levels within the L3 cells but to low levels in the L1 and L2 layers”
However, there is no evidence that HAM1 and HAM2 proteins that are synthesized in the deeper L3 layers reach the L1 layer to form protein gradients. See the work presented in (Han et al., Front. Plant Sci., Volume 11 2020. https://doi.org/10.3389/fpls.2020.541968) where HAM proteins are not detected in the L1 layer. It is an important point which shows that these proteins must act non-cell autonomously to effect CLV3 expression in the CZ. This notion of non-cell autonomous function is also suggested by original study that discovered HAM in petunia (Stuurman et al., genes and dev 2002, 16(17):2213-8) where differentiating cells thought to influence meristem maintenance. It is important to highlight this strong evidence for their non-cell autonomous function which is useful for future studies in explaining of how the WUS-HAM heterodimers that form in the inner layers influence CLV3 expression and stem cells in outer cell layers. I can also provide one more instance of discrepancy that exists in the literature which is worth highlighting to bring clarity to their non cell autonomous function. Schulze, S., et al. Plant Journal showed that only ham1;ham2;ham3 triple mutants develop extremely flattened reorganized meristems and the ham1;ham2 double mutants were closer to the wild type suggesting again that the PZ-expressed Ham3 contributes non cell autonomously. A similar conclusion on triple mutant phenotype was reached by another study Engstrom et al., Plant Physiol 2011 Feb;155(2):735-50. Highlighting these aspects is necessary to get anyway closer to the understanding the function of WUS-HAM heterodimers.
- We appreciate these comments and have eliminated the reference to a HAM concentration gradient on line 592. We also added a sentence describing the non-cell autonomous function of the HAM proteins on lines 591-592, although due to the length of the article we have not gone into all of the details provided above.
- In paragraph starting with line 602 authors discuss WUS-STM heterodimers directly activating the CLV3 expression based on Su et al., PNAS 2020.
WUS-STM heterodimers bind to upstream elements TAAT located at -1080 and STM by itself binds TGACA located at (–797 bp to –793 bp). The double mutants have been shown to abolish binding and such mutant promoter failed completely in rescuing clv3 null mutants showing their critical role. However, authors overlook mentioning two important studies. First, -1080 was identified as a WUS-binding element by Yadav et. Al., 2011 Gene and Dev, and the deletion of this element did not influence CLV3 expression. Moreover, Muller at al., Plant Cell, Volume 18, Issue 5, May 2006, Pages 1188–1198, carried out sequential deletions of the promoter. The deletions that leave behind only upstream 500bps (-500) was sufficient to fully rescue clv3 null mutants. The functionality was only compromised upon deletion of the 3’ region. These two studies contradict the critical requirement of the STM binding sites located upstream of the 500bps. These differences should be mentioned.
- We agree that although the Su et al., 2020 paper provides evidence for the importance of the promoter elements at -1080 and -797 base pairs, the other two studies mentioned contradict these findings. However, we think that the contradictory data do not invalidate the overall findings of Su et al. nor their model, and that this is too detailed a point to discuss in a general plant stem cell review. Thus we prefer to leave the paragraph as it is.
- In paragraph starting with line 649, authors discuss CK receptor requirement for activating WUS transcription. They should also mention a more direct experiment showing no effect on the WUS expression domain in triple mutants of the cytokinin receptors, rather the WUS protein stability is compromised in these mutants [Snipes et al., Plos Genetics 2018, 14(4)].
- We have added this to lines 685-686 in the paragraph that directly discusses CK regulation of WUS activity.